# Structure of the RBM7–ZCCHC8 core of the NEXT complex reveals connections to splicing factors

Sebastian Falk[1,*], Ksenia Finogenova[1,*], Mireille Melko[2], Christian Benda[1], Søren Lykke-Andersen[2], Torben Heick Jensen[2] & Elena Conti[1]

The eukaryotic RNA exosome participates extensively in RNA processing and degradation. In human cells, three accessory factors (RBM7, ZCCHC8 and hMTR4) interact to form the nuclear exosome targeting (NEXT) complex, which directs a subset of non-coding RNAs for exosomal degradation. Here we elucidate how RBM7 is incorporated in the NEXT complex. We identify a proline-rich segment of ZCCHC8 as the interaction site for the RNA-recognition motif (RRM) of RBM7 and present the crystal structure of the corresponding complex at 2.0 Å resolution. On the basis of the structure, we identify a proline-rich segment within the splicing factor SAP145 with strong similarity to ZCCHC8. We show that this segment of SAP145 not only binds the RRM region of another splicing factor SAP49 but also the RRM of RBM7. These dual interactions of RBM7 with the exosome and the spliceosome suggest a model whereby NEXT might recruit the exosome to degrade intronic RNAs.

[1] Department of Structural Cell Biology, Max-Planck-Institute of Biochemistry, Am Klopferspitz 18, D-82152 Martinsried, Germany. [2] Department of Molecular Biology and Genetics, Aarhus University, C.F. Møllers Alle 3, 8000C Aarhus, Denmark. * These authors contributed equally to this work. Correspondence and requests for materials should be addressed to T.H.J. (email: thj@mbg.au.dk) or to E.C. (email: conti@biochem.mpg.de).

The RNA exosome was first discovered from biochemical and genetic experiments in *Saccharomyces cerevisiae* as the exoribonuclease complex generating the mature 3′-end of nuclear 5.8S ribosomal RNA (rRNA)[1]. Since then, work from many laboratories has converged on the notion that the exosome is a major 3′–5′ exo- and endo-ribonuclease that degrades a bewildering number and variety of cellular transcripts in RNA processing, turnover and surveillance pathways, in the nucleus and the cytoplasm alike, and in all eukaryotes studied to date (reviewed in refs 2–6). The exosome functions together with accessory factors, which have so far been characterized primarily in *S. cerevisiae*. Crucial among these are ATP-dependent RNA helicases, which are believed to disentangle ribonucleoprotein complexes and thread unwound RNAs into the exosome channel[7]. In the cytoplasm, the Ski2 helicase associates with Ski3 and Ski8 to form the so-called Ski complex, which is recruited to the exosome core via Ski7 (refs 8–12). In the nucleus, the related helicase Mtr4 is recruited to the exosome via its interaction with Rrp6–Rrp47 and likely also Mpp6 (ref. 13). In addition, Mtr4 can separately bind the poly(A)-polymerase Trf4 and the Zinc-knuckle protein Air2 to form the Trf4/Air2/Mtr4 polyadenylation (TRAMP) complex[14–17]. TRAMP adds an oligoadenylated tail/extension to the 3′-end of specific substrates, promoting their exosomal degradation[14–16,18]. The activities of TRAMP are required for degrading defective tRNAi[Met] (ref. 19) and cryptic unstable transcripts[16,20], and are also involved in the degradation of rRNAs and small nuclear and small nucleolar RNAs[16,21–23].

An emerging general principle is that the exosome core is ubiquitously localized, whereas its accessory factors are organized in different cellular compartments. In *S. cerevisiae*, at least two versions of the TRAMP complex exist that are preferentially localized in the nucleoplasm (TRAMP4) or in the nucleolus (TRAMP5)[24,25]. Such spatial compartmentalization is even more pronounced in human cells, where the diversity of involved factors is increased. In particular, human Mtr4 (hMTR4, also known as SKIV2L2) is found in the nucleolus together with the human orthologues of the TRAMP complex (TRF4-2 and the Air2-like protein ZCCHC7), which functions in the 3′-adenylation of rRNA products[26,27]. In the nucleoplasm, however, hMTR4 binds the RBM7 and ZCCHC8 proteins, forming the metazoan-specific nuclear exosome targeting (NEXT) complex[27]. NEXT promotes the exosomal degradation of non-coding promoter-upstream transcripts[27], enhancer RNAs[28] and 3′-extended products of histone- and small nuclear RNA transcription[27,29,30]. Interestingly for the present study, it also targets intronic RNA for decay and/or processing of embedded small nucleolar RNAs[31]. An important RNA-binding element in the NEXT complex is the RNA-recognition motif (RRM) of RBM7, which shows some preference for poly-pyrimidine sequences[29,31]. This preference *in vitro* correlates with the presence of uridine-rich stretches in the targets of RBM7 in cells, although the protein is also loaded rather promiscuously on early transcripts[29,31]. RBM7/NEXT interacts not only with the exosome but also with proteins (ZC3H18 and ARS2) that connect it to the nuclear cap-binding complex, which resides on the 5′-cap structures of nuclear RNA polymerase II-derived transcripts[30,32]. While this presumably explains the moderate cap-proximal nature of RBM7 binding to RNA in cells, it has remained unclear how RBM7/NEXT gets specifically targeted to the 3′-ends of introns[31]. In this work, we started to dissect the architecture of the NEXT complex and in doing so we identified an interaction network possibly explaining how the NEXT-exosome machinery gets recruited to intronic RNA.

## Results

### ZCCHC8 is the scaffolding subunit of the NEXT complex.

The three subunits of the human NEXT complex have a multi-domain arrangement (Fig. 1a). hMTR4 (1,042 residues) harbours the same domain architecture as yeast Mtr4, with a helicase domain that contains a characteristic insertion, or arch domain, and an N-terminal low-complexity region that contains a conserved Rrp6–Rrp47-binding motif[13]. RBM7 (266 residues) features a conserved N-terminal RRM domain of about 90 amino acids[27,29] followed by a poorly conserved C-terminal region predicted to be unstructured. In the case of ZCCHC8 (707 residues), a Zinc-knuckle domain (residues 222–246) and a proline-rich region (residues 287–334) can be predicted by sequence analysis. To map the protein–protein interactions within the NEXT complex, we expressed glutathione-*S*-transferase (GST)-tagged hMTR4 and ZCCHC8 in bacterial cells and carried out pull-down assays with independently expressed Z-tagged RBM7 and ZCCHC8. Since full-length RBM7 precipitated during purification, we used a truncated fragment of RBM7 (residues 1–137 or RBM7[1–137]) lacking part of the unstructured C-terminal region of the protein. We found that RBM7[1–137] co-precipitated with GST-ZCCHC8, but not with GST-hMTR4 (Fig. 1b, lanes 7 and 8). Instead, GST-hMTR4 was able to independently precipitate Z-tagged ZCCHC8 (Fig. 1b, lane 6). We conclude that ZCCHC8 is a scaffolding subunit that bridges the interaction between RBM7 and hMTR4.

### The proline-rich segment of ZCCHC8 binds to the RBM7 RRM.

We proceeded to narrow down the interacting regions of RBM7 and ZCCHC8 through iterative rounds of construct design, pull-down assays and limited proteolysis experiments. We found that a GST-tagged version of ZCCHC8, encompassing residues 1–337, precipitated RBM7[1–137] (Fig. 1c, lane 5). RBM7[1–137] also co-precipitated with GST-ZCCH8[41–337] (Fig. 1c, lane 6), demonstrating that the interaction does not require the variable N-terminal segment of ZCCHC8. However, RBM7[1–137] did not co-precipitate with GST-ZCCHC8[1–263] (Fig. 1c, lane 7), suggesting that the segment following the Zinc-knuckle domain of ZCCHC8 is essential for the interaction.

Next we purified the complex between ZCCHC8[41–337] and RBM7[1–137], and subjected it to limited proteolysis. We observed smaller stable fragments of the two proteins. This prompted us to design new ZCCHC8 and RBM7 constructs, encompassing the RRM domain of RBM7 and the segment downstream of the Zinc-knuckle domain of ZCCHC8 (Supplementary Fig. 1a). We tested the corresponding constructs in pull-down assays and demonstrated that GST-tagged ZCCHC8[273–337] precipitated RBM7[1–98] (Fig. 1d, lane 5). This complex, however, failed to give diffraction-quality crystals. Another round of limited proteolysis and mass spectrometric analysis of the purified ZCCHC8[273–337]–RBM7[1–98] complex narrowed down the interacting regions further (Supplementary Fig. 1b). From the pull-down assays with the corresponding protein constructs (Fig. 1d, lane 6), we could demonstrate that ZCCHC8[285–324] (encompassing the proline-rich region downstream of the Zinc-knuckle domain, ZCCHC8[Pro]) is sufficient to bind RBM7[1–86] (spanning the RRM domain, RBM7[RRM]).

### Crystal structure of a ZCCHC8–RBM7 core complex.

The complex between ZCCHC8[Pro] and RBM7[RRM] yielded crystals diffracting to about 2.85 Å resolution, with seven complexes in the asymmetric unit. Molecular replacement using the nuclear magnetic resonance (NMR) structure of RBM7[RRM] (ref. 29) (PDB 2M8H) or other RRM domains (RBM11, 65% sequence identity, PDB 2YWK) as search models was not successful.

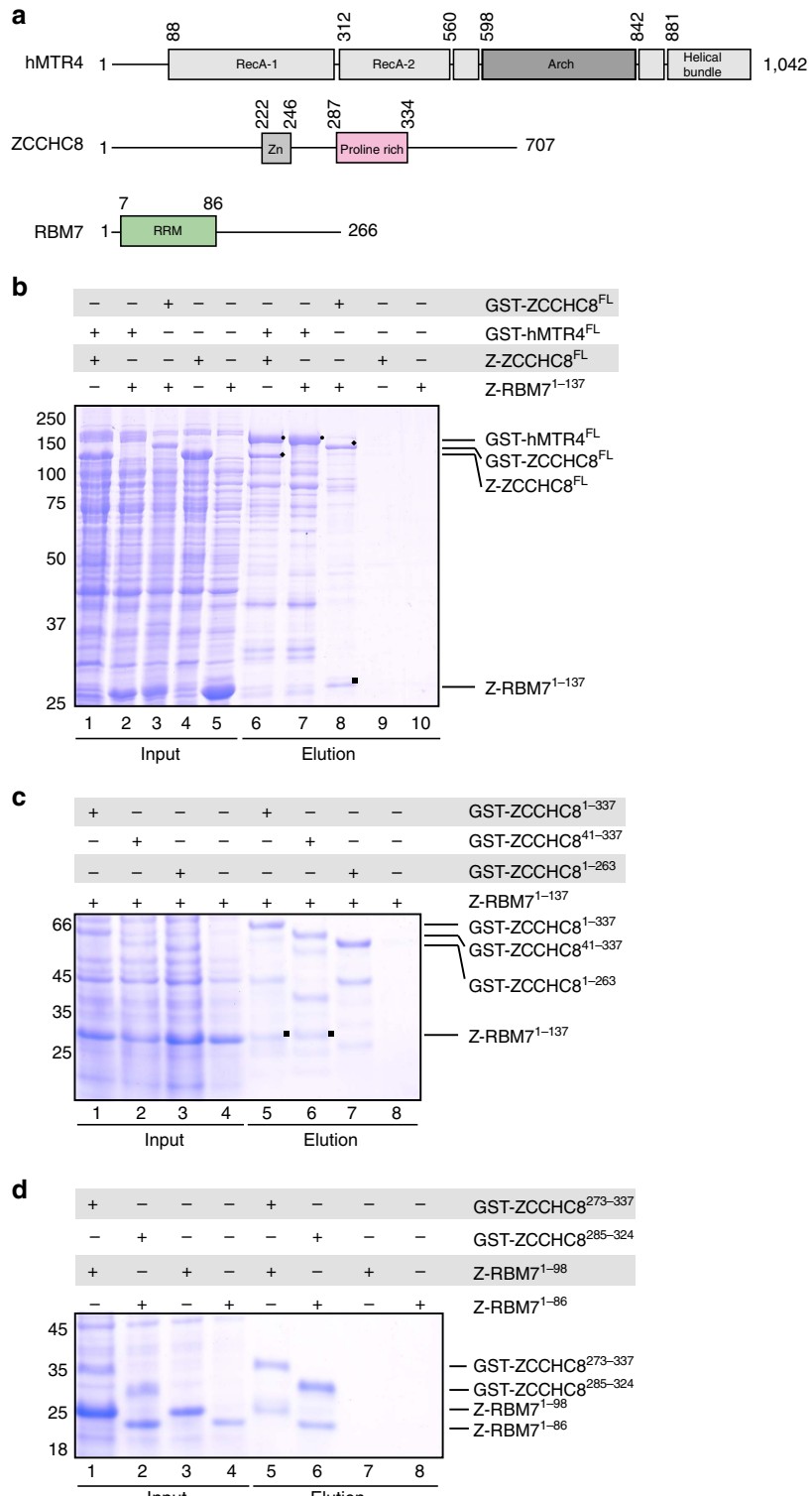

**Figure 1 | Mapping of interactions within the NEXT complex.** (**a**) Domain organization of the three subunits of the human NEXT complex. Folded domains are represented as rectangles and labelled. Predicted unstructured regions are shown as lines. The portions of RBM7 and ZCCHC8 included in the presented crystal structure are coloured in green and pink, respectively. (**b**) Protein co-precipitations by GST pull-down assays. GST-tagged ZCCHC8 or hMTR4 and Z-tagged RBM7 and ZCCHC8 were expressed individually and cells were mixed before lysis. Pull-down assays were carried out using GSH-Sepharose beads in a buffer containing 500 mM NaCl. The Coomassie-stained 12% SDS–PAGE gels show the total lysate control (lanes 1–5) and the pulled-down protein precipitates (lanes 6–10). Bands corresponding to hMtr4 (circles), ZCCHC8 (diamonds) and RBM7 (squares) are labelled. (**c**) Protein co-precipitations by GST pull-down assays. Truncated versions of GST-tagged ZCCHC8 were co-expressed with truncated versions of RBM7. Pull-down assays were carried out and analysed as described in **b**. Bands corresponding to RBM7 are labelled (squares). (**d**) Reconstitution of a minimal RBM7–ZCCHC8 core complex. Protein co-precipitations by GST pull-down assays. Truncated versions of GST-tagged ZCCHC8 were co-expressed with truncated versions of RBM7. Pull-down assays were carried out and analysed as described in **b**.

**Table 1 | Crystallographic data collection and refinement statistics.**

| Data set | RBM7$^{RRM}$–ZCCHC8$^{Pro}$Sm$^{3+}$ | RBM7$^{RRM}$–ZCCHC8$^{Pro}$native |
|---|---|---|
| Space group | P6$_3$22 | C2 |
| *Cell dimensions* | | |
| $a, b, c$ (Å) | 79.5, 79.5, 87.6 | 178.8, 66.6, 111.9 |
| $\alpha, \beta, \gamma$ (°) | 90, 90, 120 | 90, 126.6, 90 |
| | | |
| *Data collection* | | |
| Wavelength (Å) | 1.60 | 1.00 |
| Resolution (Å) | 87.61–2.00 | 71.79–2.85 |
| $R_{merge}$ | 18.7 (344.2) | 6.1 (114.2) |
| $I/\sigma I$ | 26.6 (1.34) | 8.1 (0.8) |
| Completeness (%) | 98.8 (92.7) | 99.8 (99.7) |
| Multiplicity | 134.8 | 3.2 |
| CC$_{1/2}$ | 100 (34.1) | 99.5 (62.0) |
| | | |
| *Refinement* | | |
| Resolution (Å) | 87.61–2.00 (2.08–2.00) | 71.79–2.85 (2.92–2.85) |
| No. of unique reflections | 11,456 | 24,933 |
| Copies per a.s.u. | 1 | 7 |
| $R_{work}/R_{free}$ (%) | 20.2/22.6 | 26.2/30.0 |
| Wilson B-factor (Å$^2$) | 43.4 | 88.7 |
| Average B-factors (Å$^2$) | 49.4 | 107.3 |
| No. of atoms | | |
| Proteins | 955 | 5,485 |
| Ligands | 9 | 9 |
| Stereochemistry | | |
| R.m.s.d. bond lengths (Å) | 0.003 | 0.005 |
| R.m.s.d. bond angles (°) | 0.56 | 1.12 |
| Ramachandran favoured (%) | 100 | 98.0 |
| Ramachandran outliers (%) | 0.0 | 0.4 |

R.m.s.d., root mean squared deviation; a.s.u., asymmetric unit.

To obtain experimental phases, we co-crystallized the complex in the presence of heavy-atom compounds. Co-crystallization with Samarium chloride yielded a different crystal form diffracting to 2.0 Å resolution and containing a single copy of the complex in the asymmetric unit. The structure was solved by single anomalous dispersion at the Samarium edge and refined to 2.0 Å resolution with $R_{free}$ of 22.6%, R-factor of 20.2% and good stereochemistry (Table 1). The coordinates of the Sm-derivatized RBM7$^{RRM}$–ZCCHC8$^{Pro}$ complex allowed solving the structure of the native complex ($R_{free}$ of 30.0% and R-factor of 26.2%; Table 1). As the two structures are very similar (Supplementary Fig. 2a), we focused on the highest-resolution atomic model, which includes residues 286–324 of ZCCH8 and residues 7–86 of RBM7. Side-chain electron density was well resolved throughout the RBM7$^{RRM}$–ZCCHC8$^{Pro}$ interface (Supplementary Fig. 2b).

**ZCCHC8 is recognized at the helical surface of the RBM7 RRM.** RBM7$^{RRM}$ folds into the typical globular domain with four antiparallel β-strands (β1–β4) at the front and two α-helices at the back (α1 and α2, encompassing residues 23–33 and 60–70, respectively; Fig. 2a). In comparison with the previously reported NMR structure of RBM7$^{RRM}$ (ref. 29), we found longer secondary structure elements for α2 and β4 and an additional β-strand in the α2–β4 loop (referred to as β4$^{add}$) (Supplementary Fig. 2c and Supplementary Fig. 2d). When viewing the structure at the front, the β4$^{add}$–β4–β1–β3–β2 sequel of strands positions short, medium and long loops at the bottom of the molecule (β4$^{add}$–β4 loop residues 75–78, β1–α1 loop residues 15–23 and β2–β3 loop residues 40–53, respectively). The conformation of these loops is restrained by the presence of proline residues and of intramolecular interactions, including main-chain hydrogen bonds and van der Waals contacts between aliphatic side chains

(Supplementary Fig. 2e). Their conformation is indeed similar to that observed in the NMR structure.

In the complex, RBM7$^{RRM}$ exposes the front β-sheet surface to solvent and engages the helical back surface in binding ZCCHC8$^{Pro}$ with conserved interactions (Fig. 2b). The interaction covers 18% of the total accessible surface area of RBM7$^{RRM}$ as calculated by the PISA server[33]. ZCCHC8$^{Pro}$ positions the N terminus at the top of the RRM and then stretches downward, laying over helix α1 and reaching the bottom of the domain. Here the polypeptide chain of ZCCHC8$^{Pro}$ makes a ~90° bend and continues laterally with an α-helix (αA, residues 293–299). ZCCHC8$^{Pro}$ then twists into a ~90° coil and continues upward with a second α-helix (helix αB, residues 308–316), reaching the top of RBM7$^{RRM}$. Finally, ZCCHC8$^{Pro}$ makes another ~90° bend and extends laterally over helix α2, ending with a short helical turn (residues 320–323). The C- and N-terminal residues of ZCCHC8$^{Pro}$ interact with each other at the top of RBM7$^{RRM}$. Overall, ZCCHC8$^{Pro}$ adopts an unusual elliptical conformation when bound to RBM7. ZCCHC8$^{Pro}$ also contains an unusually high percentage of proline residues[34] that contribute in shaping the path of the polypeptide chain due to the conformational rigidity of their cyclic structure.

**The ZCCHC8–RBM7 interface is evolutionary conserved.** The RBM7$^{RRM}$–ZCCHC8$^{Pro}$ interaction is dominated by conserved hydrophobic contacts that can be grouped into two adjacent patches (Fig. 2a,b). The first interaction hotspot (patch 1) is in the lower half of the RRM domain: ZCCHC8$^{Pro}$ helix αA together with the preceding and following loops insert a set of apolar residues into a shallow hydrophobic pocket formed between the α1 and β4$^{add}$ elements of RBM7$^{RRM}$ (Fig. 2c,d). In particular, Leu295, Ala298, Leu299 and Phe309 of ZCCHC8 are in van der

Waals contacts with Leu25, Leu29, Ile73, Leu75 and Tyr76 of RBM7. This layer of hydrophobic residues packs against an outer layer formed by ZCCHC8 Ile291, Val301, Pro307, Pro308 and Ile310. The second interaction hotspot (patch 2) is in the upper half of the RRM domain. Here helix αB as well as the C-terminal

and N-terminal segments of ZCCHC8 cluster a set of apolar side chains into a hydrophobic surface groove formed between helices α1 and α2 of RBM7 (Fig. 2e). Leu69 of RBM7 projects from helix α2 to interact with ZCCHC8 Met313, while RBM7 Tyr65 stacks against a proline–proline dipeptide in ZCCHC8 (Pro319 and

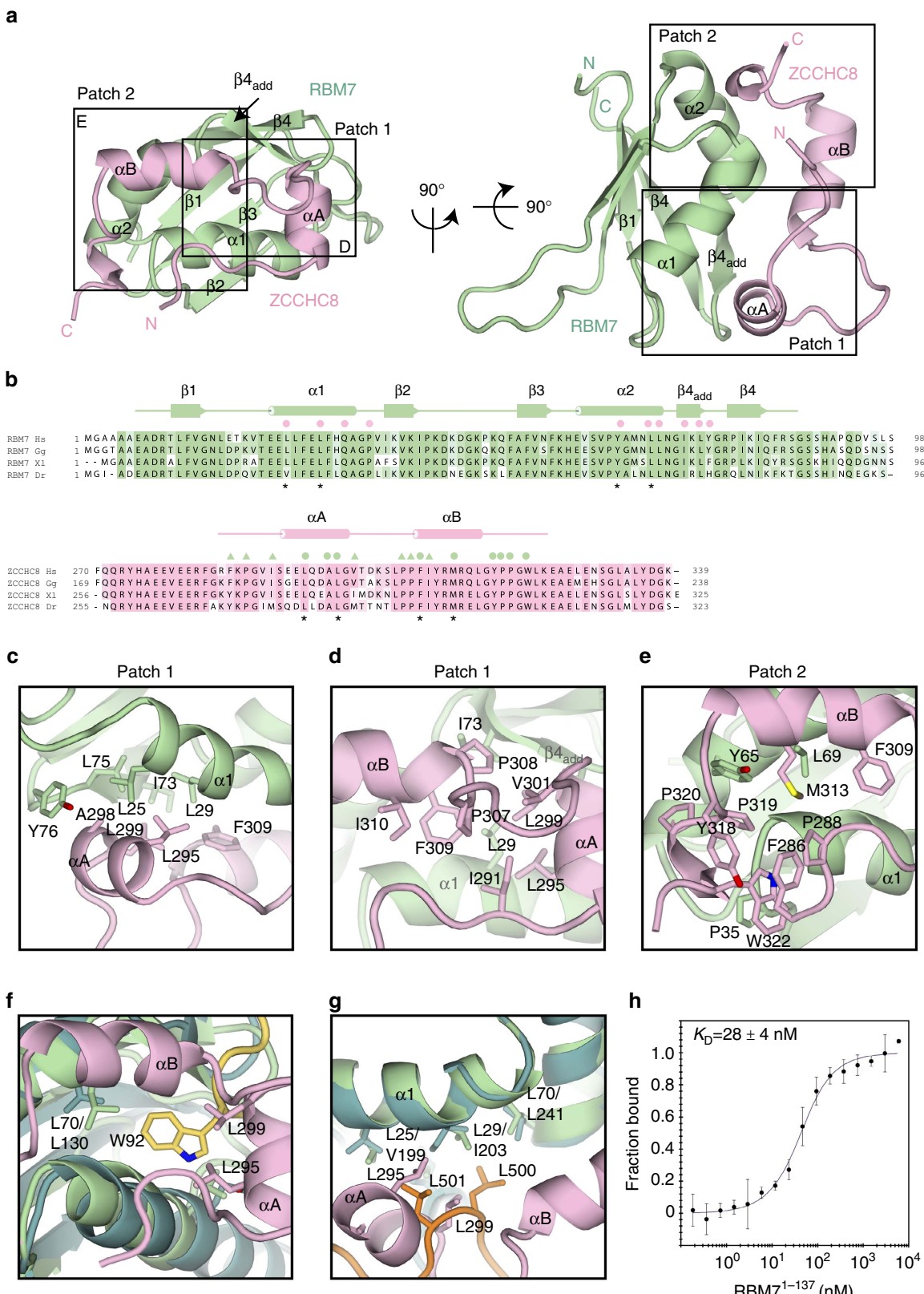

Pro320). Adjacent to it, the N- and C-terminal residues of ZCCHC8[Pro] form a remarkable array of consecutive face-to-face stacking interactions that involve Pro288, Phe286, Trp322 and RBM7 Pro35. Finally, ZCCHC8 Tyr318 and Phe286 lay on top of each other with an edge-to-face aromatic interaction.

The interactions at patch 1 are reminiscent of those observed in other RRM–protein complexes. For example, ZCCHC8 Leu295 and Leu299 are at a similar structural position as a tryptophan residue found in motifs that recognize the UHM (U2AF homology motif) family of RRM[35–37] (Fig. 2f) and also observed in the interaction of eIF3j with the RRM of eIF3b (ref. 38; Supplementary Fig. 2f). Even more striking is the similarity with the complex between the polypyrimidine tract-binding protein (PTB) and Raver1 (ref. 39). Raver1 inserts a pair of leucine residues (corresponding to ZCCHC8 Leu295 and Leu299) into an equivalent hydrophobic pocket in the PTB RRM that has very similar surface features as RBM7 (Fig. 2g). U2AF[65] and U2AF[35] (UHM–peptide complex) form a very tight complex with a dissociation constant of $K_D \sim 2$ nM (ref. 35), whereas in comparison the PTB–Raver1 interaction is rather weak, with a $K_D \sim 100 \mu$M (ref. 40). To test in which affinity regime the RBM7[RRM]–ZCCHC8[Pro] complex resides, we determined the dissociation constant by microscale thermophoresis. We engineered and purified a ZCCHC8[Pro]-(GS)$_3$-eYFP fusion protein (using a Gly–Ser linker). Titration of ZCCHC8[Pro]-(GS)$_3$-eYFP with RBM7[1–137] yielded a $K_D$ of 28 nM ($\pm 4$ nM) (Fig. 2h), suggesting that RBM7[RRM]–ZCCHC8[Pro] form a tight complex similar to that of U2AF[65] and U2AF[35]. Finally, the RBM7[RRM]–ZCCHC8[Pro] interaction does not use the front surface of the RRM domain, which is expected to mediate RNA binding. Indeed, the RBM7[RRM]–ZCCHC8[Pro] complex was able to bind a polyU RNA substrate in fluorescence anisotropy experiments and displayed similar RNA-binding affinity as compared with RBM7 in isolation (Supplementary Fig. 2g).

**Mutational analysis of the ZCCHC8–RBM7 interface**. We next tested experimentally the importance of the hydrophobic contacts observed in the crystal structure. To this end, specific amino-acid substitutions in RBM7[RRM] and ZCCHC8[Pro] were engineered and their impact on the RBM7–ZCCHC8 interaction was tested by co-expression pull-down assays *in vitro*. In these assays, RBM7[RRM] patch 2 substitutions (Tyr65Ala and Leu69Glu) or ZCCHC8[Pro] substitutions (Met313Glu and Phe309Ala) severely weakened the interaction of the polypeptides (Fig. 3a, compare lane 8 with lanes 10 and 12). An even more drastic effect was observed when mutating patch 1 either in RBM7[RRM] (Leu25Glu and Leu29Glu) or in ZCCHC8[Pro] (Leu295Glu and Leu299Glu), which essentially abrogated complex formation (Fig. 3a, lanes 9 and 11).

To evaluate whether the patch 1 mutations would also affect RBM7–ZCCHC8 interaction in cells in the background of the full-length proteins, we carried out immunoprecipitation (IP) experiments using extracts from HEK293 cells overexpressing full-length wild-type (WT) or patch 1 mutant (mut) versions of RBM7-GFP (Fig. 3b, left panels) or ZCCHC8-FLAG (Fig. 3b, right panels) fusion proteins. Using the engineered affinity tags, both WT and mut proteins were immunoprecipitated efficiently (Fig. 3b). Moreover, consistent with our *in vitro* experiments only WT RBM7-GFP and ZCCHC8-FLAG were able to IP endogenous ZCCHC8 and RBM7, respectively, whereas their mut counterpart proteins failed to do so (Fig. 3b, compare lanes 5 and 6 for both left and right panels).

**Binding of ZCCHC8 and SAP145 to RBM7 is mutually exclusive**. The identification of the proline-rich RRM-binding motif in ZCCHC8 prompted us to ask whether other proteins might harbour a similar functional site. In bioinformatic analyses, we detected a remarkable sequence similarity between the proline-rich ZCCHC8 segment and a proline-rich stretch in the protein SAP145 (residues 603–647, SAP145[Pro]) (Fig. 4a, upper panel). SAP145 (also known as SF3b2) is a subunit of the spliceosomal SF3b complex. Previous interaction studies by far-western analyses suggested the presence of a specific SF3b subcomplex between SAP145 and SAP49 (also known as SF3b4), a protein containing two RRM domains[41]. We reasoned that the interaction between SAP49 and SAP145 might resemble that of RBM7[RRM] and ZCCHC8[Pro]. SAP49 contains two RRMs at the N terminus, and sequence alignments suggest that the first one (RRM1) is the most similar to RBM7 (Fig. 4a, lower panel). Indeed, *in vitro* experiments with recombinant proteins showed that SAP145[Pro] interacts directly with a SAP49 construct comprising both RRMs (SAP49[RRM1–2]) or RRM1 only (SAP49[RRM1]) (Fig. 4b, lane 10 and 11). As a control, SAP145[Pro] did not interact with p14, the other RRM-containing protein of the SF3b complex (Supplementary Fig. 3, lane 7).

The striking similarity between ZCCHC8[Pro] and SAP145[Pro] and between RBM7[RRM] and SAP49[RRM] (in particular the first RRM) raised the possibility of an interaction across the NEXT and SF3b complexes. In support of this, mining the data from recent proteomic studies in human cells[27,30,42,43] revealed that affinity purifications of the bait protein RBM7 contained SAP145 and other SF3b subunits, in addition to ZCCHC8, hMTR4 and exosome subunits. Furthermore, RBM7 was previously reported to associate with SAP145 in rat testes[44]. We assessed the suggested interaction *in vitro*, and found that the RBM7[RRM] and SAP145[Pro] fragments make direct physical contact (Fig. 4b, lane 7). Mutation of the patch 1 surface of RBM7[RRM] had a minor effect on SAP145[Pro] binding (Fig. 4b, lane 8), whereas mutation of patch 2 effectively abolished the interaction (Fig. 4b, lane 9). Since both ZCCHC8 and SAP145 proline-rich domains bind to RBM7, we also tested if they could form a ternary complex. To this end,

**Figure 2 | Structure of the RBM7–ZCCHC8 core complex.** (**a**) Crystal structure of the RBM7–ZCCHC8 core complex shown in two orientations. RBM7 is in green and ZCCHC8 in pink. Secondary structure elements discussed in the text are indicated. Boxes indicate the regions of the RBM7–ZCCHC8 interface that are shown as zoom-ins in **c–e**. (**b**) Structure-based sequence alignments of RBM7 and ZCCHC8, including orthologues from *Homo sapiens* (Hs), *Gallus gallus* (Gg), *Xenopus laevis* (Xl) and *Danio rerio* (Dr). The secondary structure elements are shown above the sequences. Conserved residues are highlighted in green (RBM7) and pink (ZCCHC8). Above the sequences, coloured circles identify residues involved in intermolecular interactions with ZCCHC8 (pink circles) and with RBM7 (green circles). Green triangles indicate the residues that form the outer hydrophobic layer. Asterisks point to the residues mutated in the interaction studies below. (**c–e**) Zoom-in view of a representative set of residues at the RBM7–ZCCHC8 interaction patch 1 (**c,d**) and patch 2 (**e**) (see text for details). (**f,g**) Comparison of interactions at the back helical surface of RRM domains. The two panels show the superposition of RBM7–ZCCHC8 with U2AF[35]/U2AF[65] (**f**) and with PTB–Raver (**g**). The zoom-in views show how an equivalent hydrophobic pocket in the three RRM-containing proteins can accommodate either a Trp residue (**f**) or two Leu residues (**g**). The RRM domains of U2AF[35] and PTB are shown in dark green, U2AF[65] in yellow and the Raver peptide in orange. (**h**) The binding of RBM7[1–137] to ZCCHC8[Pro] was measured with microscale thermophoresis MST (circles). The titration of RBM7[1–137] ranged from 0.2 nM to 6.0 μM with a constant concentration of ZCCHC8[Pro]-(GS)$_3$-eYFP at 30 nM. Data were analysed by temperature jump mode and yielded a $K_D$ of 28 ± 4 nM. The error bars represent the s.d. of each data point calculated from three independent thermophoresis measurements.

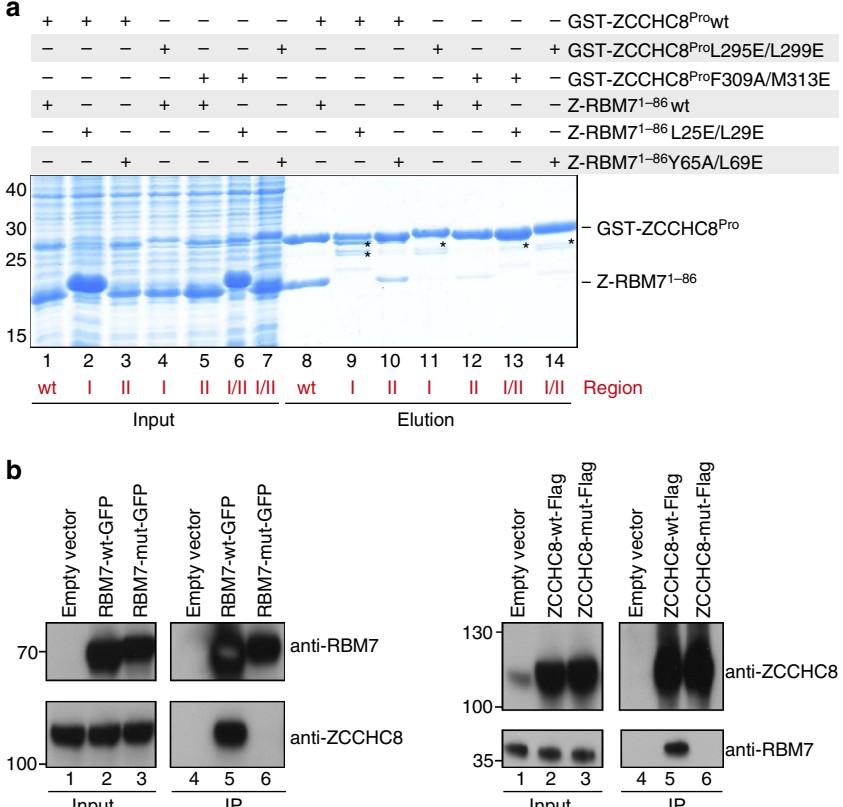

**Figure 3 | Mutational analysis of the ZCCHC8–RBM7 interface.** (**a**) Effect of structure-based mutations *in vitro* assayed by protein co-precipitations in GST pull-down assays. WT and indicated mutant versions of GST-tagged ZCCHC8 were co-expressed with WT and indicated mutant versions of RBM7. Pull-down assays were carried out and analysed as described in Fig. 1b. The mutants map to patches I and II (as indicated in red at the bottom of the gels). Asterisks point to ZCCHC8 degradation products, which arose when the protein was absent from RBM7. (**b**) Patch 1 facilitates RBM7–ZCCHC8 interaction in cells. Western blotting analysis of input and RBM7-GFP or ZCCHC8-FLAG IP samples. Full-length WT or patch 1 mutant (mut) proteins were expressed transiently in HEK293 cells. An empty expression vector was used as a negative control. IPs were carried out using Dynabeads coupled to anti-GFP nanobodies or to anti-FLAG antibodies. Input and IP fractions were probed with anti-RBM7 and anti-ZCCHC8 antibodies as indicated.

we co-expressed MBP-RBM7$^{1-86}$ with GST-ZCCHC8$^{Pro}$ and TRX-SAP145$^{Pro}$ and performed two pull-down experiments. When using MBP-binding amylose beads, MBP-RBM7$^{1-86}$ co-precipitated both GST-ZCCHC8$^{Pro}$ and TRX-SAP145$^{Pro}$ (Fig. 4b, lane 13). However, when using GST-binding GSH beads, GST-ZCCHC8$^{Pro}$ precipitated only MBP-RBM7$^{1-86}$ (Fig. 4b, lane 12). Therefore, the interaction of the proline-rich domains of ZCCHC8 and SAP145 with RBM7$^{1-86}$ appears to be mutually exclusive. This supports the notion that the back helical surface of RBM7$^{RRM}$ can recognize the proline-rich segment of either ZCCHC8$^{Pro}$ or SAP145$^{Pro}$. Detailed differences in the amino-acid sequences of ZCCHC8 and SAP145 might be responsible for tighter binding at the patch 1 surface of RBM7$^{RRM}$ (in the case of ZCCHC8$^{Pro}$) or patch 2 (in the case of SAP145$^{Pro}$).

The interaction between RBM7 and an SF3b protein raised the question whether NEXT plays a role in splicing. To address this question, we quantitatively analysed changes in alternative splicing using the MISO algorithm[45] on RNA-seq data sets obtained from HeLa cells depleted for either RBM7 or ZCCHC8, and two corresponding control samples[30] (Supplementary Fig. 4a–c). We identified only few consistently changed alternative splicing events: 14 in RBM7-depleted samples (Supplementary Fig. 4a,d) and 32 in ZCCHC8-depleted samples (Supplementary Fig. 4b,d). Only four of these altered splicing events were observed in both RBM7 and ZCCHC8 depletions (Supplementary Fig. 4d). Thus, neither RBM7 nor ZCCHC8 appear to function as general splicing factors. Previous, iCLIP

data have shown that RBM7 binds to a region in the 3′-ends of introns, at a similar position where the U2snRNP binds[31]. In addition, SAP49 has been shown to crosslink near the branch site sequence in the splicing reaction[41]. Altogether, these findings suggest a model where the interaction with SF3b could help recruit NEXT to introns to assist in RNA degradation after completion of the splicing process.

## Discussion

In this work, we have shown that RBM7$^{RRM}$ binds a proline-rich segment of ZCCHC8 at the back helical surface and is able to concomitantly bind RNA. This RBM7$^{RRM}$–ZCCHC8$^{Pro}$ interaction engages extensive surfaces and thus appears to be a stable architectural unit in the formation of the NEXT complex. On the basis of this structural analysis, we found that RBM7$^{RRM}$ can interact with SAP145$^{Pro}$, a homologous proline-rich segment present in the spliceosomal SF3b complex. In turn, SAP145$^{Pro}$ binds the RRM region of the splicing factor SAP49 (ref. 44), linking RBM7 to RNA splicing factors, in addition to its connection to exosome-mediated RNA degradation[27,30,46]. The mutually exclusive binding of ZCCHC8 and SAP145 to RBM7 could reflect independent functions of RBM7 in exosome-mediated degradation and splicing. Alternatively, it could reflect the ability of concomitantly linking both machineries provided RBM7 is able to homo-dimerize as reported for RBM11, a paralogue of RBM7 that also associates

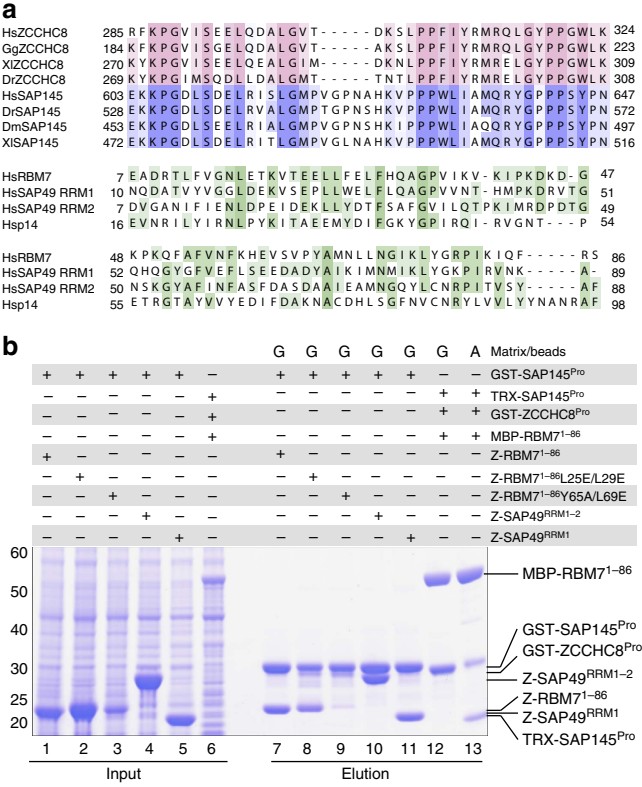

**Figure 4 | RBM7 forms alternative interactions with splicing factors.** (**a**) Upper panel: sequence alignments of the RBM7-binding region of ZCCHC8 proteins from the indicated species (from Fig. 2b) with their respective SAP145 orthologues. ZCCHC8[Pro] is shown in pink and SAP145[Pro] in blue. Lower panel: sequence alignment of the ZCCHC8-binding RRM of RBM7 with RRM1 and RRM2 of SAP49 and the RRM of p14. (**b**) The proline-rich domains of ZCCHC8 and SAP145 bind to RBM7 in a mutually exclusive manner. GST-tagged ZCCHC8[Pro] or SAP145[Pro] were co-expressed with truncated versions of RBM7 WT or mutants and SAP49. Pull-down assays were carried out and analysed as described in Fig. 1b. The resin used is indicated above the lane: MBP-binding amylose resin (A) and GST-binding GSH resin (G).

with ZCCHC8 and SAP145 (ref. 42), and has a preference for uridine-rich sequences[47]. Regardless the exact details, this interaction network together with previous iCLIP data[31] suggests a mechanism by which the NEXT–exosome machinery can be targeted to the 3′-ends of introns. While NEXT does not have a general effect on splicing, we speculate that the coupling between NEXT and SF3b complexes might serve in targeting NEXT to intronic RNA for degradative purposes.

## Methods

**Protein purification.** His-GST-tagged ZCCHC8[285–324] was co-expressed with His-Z-tagged RBM7[1–86] in bacterial cells. The complex was purified by a glutathione affinity step. The tags were cleaved by 3C and removed by a Ni-nitrilotriacetate affinity step. In the final purification step, the complex was subjected to size-exclusion chromatography in 20 mM Tris/HCl (pH 7.5), 150 mM NaCl and 2 mM dithiothreitol.

**Crystallization and structure determination.** After size-exclusion chromatography, the complex was concentrated to 20 mg ml$^{-1}$ and stored at 4 °C until further use. Crystallization trials were performed using a vapour diffusion set-up by mixing the protein complex and crystallization solution in a 2:1 ratio. Initial native crystals grew in 0.1 M Bis-Tris-Propane (pH 6.5), 0.2 M NaBr and 20% (w/v) PEG3350 (PACT Suite screen, F2, Qiagen) at 18 °C. The initial native hit was refined to 0.1 M Bis-Tris-Propane (pH 6.5), 0.2 M NaBr, 0.1 M sodium malonate and 20% (w/v) PEG3350 (Additives Screen, Hampton Research, C1). The crystals were cryoprotected with the reservoir solution supplemented with 20% (w/v) ethylene glycol before data collection at 100 K. All diffraction data were

collected at the Swiss Light Source PXII beamline (Villigen, Switzerland). The data of the Samarium derivative crystals were processed using XDS[48], and the data of the native crystals were processed with Xia2/Dials[49] within CCP4i2 (ref. 50). Native crystals belong to the monoclinic spacegroup C2 and diffracted to 2.85 Å resolution. For co-crystallization with Samariumchloride, the protein complex was mixed with SmCl$_3$ dissolved in buffer in a 1:2 molar ratio. Crystals grew in 0.1 M Bis-Tris-Propane (pH 6.5), 0.2 M NaBr, 18% (w/v) PEG3350 and 7% (w/v) glycerol at 18 °C, and were cryoprotected with the reservoir solution supplemented with 20% (w/v) glycerol before data collection at 100 K. Samarium derivatized crystals belong to the hexagonal spacegroup P6$_3$22 and diffracted to 2 Å resolution. The structure of ZCCHC8[Pro] and RBM7[RRM] was solved by single anomalous dispersion phasing with Autosol from Phenix[51]. The mean figure of merit over all resolution shells had a value of 0.46 and estimated map correlation coefficient a value of 63.3 ± 6.9. Most of the model was automatically built using Buccaneer[52], was manually completed with COOT[53] and refined with phenix.refine[54]. The obtained model was used to solve the structure of the native crystals by molecular replacement with Phaser within Phenix[55]. The native crystals contain seven RBM7[RRM]–ZCCHC8[Pro] complexes (14 chains) in the asymmetric unit. The model was manually completed with COOT and refined with phenix.refine[54].

**Pull-down assays.** For interaction studies the respective combination of proteins were either co-expressed or individually expressed and co-lysed in 20 mM Tris/HCl (pH 7.5), 500 mM NaCl, 10 mM imidazole, 2.5 mM EDTA, 10% (v/v) glycerol, 0.1% (v/v) NP40, 5 mM β-mercaptoethanol and 1 mM phenylmethylsulfonyl fluoride. ZCCHC8[273–337] and ZCCHC8[285–324] were co-expressed with RBM7[1–98] and RBM7[1–86], respectively, to avoid degradation of ZCCHC8. Glutathione sepharose beads (GE Healthcare) were incubated with lysates for 2 h, washed three times with 0.5 ml lysis buffer and the retained material was eluted with lysis buffer containing 30 mM glutathione. Input material and eluates were analysed by SDS–PAGE and Coomassie staining.

Uncropped SDS–PAGE gels are shown in Supplementary Fig. 5.

**Interaction studies in cells.** To construct patch 1 'Mut' versions of ZCCHC8-FLAG and RBM7-GFP, pcDNA5-ZCCHC8-FLAG and pcDNA5-RBM7-GFP plasmids[27] were subjected to site-directed mutagenesis using the following oligonucleotide primer pairs:

RBM7-GFP-Mut:
5′-GTGACCGAGGAGGgaaCTTTTCGAGgaaTTCCACCAGGCTGG-3′
5′-CCAGCCTGGTGGAAttcCTCGAAAAGttcCTCCTCGGTCAC-3′

ZCCHC8-FLAG-Mut:
5′-GGAGTTATTAGTGAGGGAAgaaCAAGATGCAgaAGGTGTGACAGAC-3′
5′-GTCTGTCACACCTtcTGCATCTTGttcTTCCTCACTAATAACTCC-3′

Plasmids were transfected into HEK293 cells (Flp-In 293 T-REx cells, Thermo Scientific: R78007), maintained in Dulbecco's modified Eagle's medium supplemented with 10% fetal bovine serum (Invitrogen), using the calcium-phosphate method. Cells were transfected with 20 µg of plasmid DNA mixed in 500 µl of 0.25 M CaCl$_2$ and 500 µl of 2 × HEBS buffer (250 mM NaCl, 9 mM KCl, 1.5 mM Na$_2$HPO$_4$, 10 mM glucose and 50 mM HEPES/NaOH, pH 7.1), collected after 48 h and lysed in lysis buffer (200 mM NaCl, 20 mM HEPES/NaOH (pH 7.4) and 0.5% Triton X-100) containing protease inhibitor cocktail (Roche). Cell lysates were sonicated 3 × 10 s using 20 W and cleared by centrifugation at 4,000g for 15 min at 4 °C. IPs using equal amounts of lysates were performed using anti-Flag M2 antibodies (Sigma) or anti-GFP nanobodies coupled to Epoxy Dynabeads M-270 beads for 1 h in lysis buffer at 4 °C. After washing of beads three times with lysis buffer, proteins were eluted with NuPAGE LDS sample buffer (Life Technologies) at 75 °C for 10 min and denatured by adding NuPAGE Sample Reducing Agent (Life Technologies) at 75 °C for 10 min. One per cent of initial input material and 25% of eluate were run on NuPAGE Novex 4–12% Bis-Tris Protein Gels (Life Technologies). After transfer to membranes, western blotting analysis was performed according to standard procedures using the following primary antibodies: polyclonal rabbit anti-RBM7 (1: 1,000, Sigma: HPA013993); and monoclonal mouse anti-ZCCHC8 (1:2,000, Abcam: ab68739). Anti-mouse and anti-rabbit IgGs conjugated with horseradish peroxidase (Dako) were used as secondary antibodies. Uncropped western blots are shown in Supplementary Fig. 5.

**Microscale thermophoresis.** To determine the dissociation constant of the RBM7–ZCCHC8 complex, 30 nM of ZCCHC8[Pro]-(GS)$_3$-eYFP was incubated with increasing concentrations of unlabelled RBM7[1–137] in 50 mM Tris/HCl (pH 7.4), 150 mM NaCl, 10 mM MgCl$_2$ and 0.05% (v/v) Tween 20. The RBM7[1–137] concentration series was produced by serial dilution (1:1). Thermophoresis was measured with an LED power of 70% and standard parameters on a NanoTemper Monolith NT.115 machine. Titrations were performed in triplicates and the data were analysed using the T-Jump strategy option with the MO software (Nano-Temper Technologies).

**Fluorescence anisotropy.** Fluorescence anisotropy measurements were performed with a 5′-6-carboxy-fluorescein (6-FAM)-labelled U8 RNA at 20 °C in 50 µl reactions on a Genios Pro (Tecan). The RNA was dissolved to a concentration of 10 nM and

# ARTICLE

incubated with RBM7 complexes at different concentrations in a buffer containing 20 mM Tris/HCl (pH 7.5), 150 mM NaCl and 2 mM dithiothreitol. The excitation and emission wavelengths were 485 and 535 nm, respectively. Each titration point was measured 3 times using 10 reads with an integration time of 40 μs. The data were analysed by nonlinear regression fitting using the BIOEQS software.

**Analysis of RNA-seq data.** RNA-seq data from HeLa cells subjected to control (× 2), RBM7 or ZCCHC8 depletion using siRNA-mediated knockdown were obtained from the Sequence Read Archive database (accession code SRP031620). The mapped reads[30] were used to assess alternative splicing by use of the MISO algorithm and annotated human splicing events (Human genome (hg19) alternative events v2.0 downloaded from the MISO homepage http://miso.readthedocs.io/en/fastmiso/annotation.html#version-2-alpha-release-of-the-human-mouse-annotations-compiled-june-2013)[45]. Significant altered splicing events were determined by requiring a Bayes factor $\geq 10$, $|\Delta\Psi| \geq 0.10$, number of reads supporting the first isoform $\geq 1$, number of reads supporting the second isoform $\geq 1$ and the sum of reads supporting both isoforms $\geq 10$. For both RBM7 and ZCCHC8 knockdown samples, consistent altered splicing events were determined by finding the overlap between significant events compared with two different control samples (ctrl1 and ctrl2), and asserting that the direction of splicing change was consistent between the two comparisons. Finally, the overlap between consistent significant events for RBM7 and ZCCHC8 knockdown samples were determined.

**Data availability.** The coordinates and the structure factors have been deposited in the Protein Data Bank with accession codes 5LXR (Sm$^{3+}$ derivative) and 5LXY (native).

The RNA-seq data with the accession code SRP031620 were retrieved from the Sequence Read Archive database.

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

## Acknowledgements

We thank Jérôme Basquin, Karina Valer-Saldaña and Sabine Pleyer at the MPI-Martinsried crystallization facility, and Michal Lubas for intellectual input at the early phase of the project. We thank the staff of the PX beamlines at the Swiss Light Source (Villigen, Switzerland) for assistance during data collection, Claire Basquin for fluorescence anisotropy measurements, Timon Nast-Kolb for help with protein purifications and microscale thermophoresis measurement during his internship and Andrew MacMillan for the DNA encoding p14. This study was supported by the Max Planck Gesellschaft, the European Commission (ERC Advanced Investigator Grant 294371 and Marie Curie ITN RNPnet) and the Deutsche Forschungsgemeinschaft (DFG SFB646, SFB1035, GRK1721, FOR1680 and CIPSM) to E.C. and by the Danish Cancer Society and the European Commission (ERC Advanced Investigator Grant 339953) to T.H.J.

## Author contributions

K.F. solved the structure with help from S.F. and C.B.; S.F. and K.F. carried out the *in vitro* experiments; M.M. carried out the experiments in cells; S.L.-A. analysed RNAseq data; S.F., E.C. and T.H.J. initiated the project; E.C., S.F., K.F. and T.H.J. wrote the manuscript.

## Additional information

**Competing financial interests:** The authors declare no competing financial interests.

