## [Peer Review File · Nature Communications]

Reviewers' comments:

Reviewer #1 (Remarks to the Author):

Finogenova and colleagues investigated aspects of the structural organization of the Nuclear Exosome Targeting (NEXT) complex, consisting of the RBM7, ZCCHC8 and MTR4 subunits, and present evidence for interactions between RBM7 and the splicing factor SAP145. They identified a proline-rich segment of ZCCHC8 that binds the N-terminal RNA recognition motif (RRM) of RBM7 and determined crystal structures of the complex. Based on the structures, they designed mutations in the two interaction partners that affect two neighboring hydrophobic interaction patches and that weakened or abrogated complex formation. They also demonstrate similar effects of the mutations in vivo using immunoprecipitation assays. Furthermore, they found a region of high sequence similarity to the RBM7-binding region of ZCCHC8 in the splicing factor SAP145, as well as a similarity between the RBM7 RRM and the first RRM of SAP49, a known binding partner of SAP145. Based on these findings, they demonstrate a direct interaction between the ZCCHC8-like region in SAP145 and the RBM7 RRM that is mutually exclusive with ZCCHC8 binding.

The work presented is of high technical quality and the results are interesting. The crystal structures document a novel mode of protein-RRM interaction that is apparently used by various RBM7 ligands and that leaves the canonical RNA-binding surface of the RBM7 RRM unobstructed.

However, as presented the results provide limited new insights into the function of the NEXT complex or its putative link to splicing. Given the mutually exclusive binding of ZCCHC8 and SAP145 to RBM7, which might rather argue for an independent function of RBM7 in the two contexts, the authors speculate that dimerization of RBM7 might mediate a physical connection between the NEXT complex and the spliceosome (based on the known dimerization of a RBM7 paralogue, RBM11). However, direct evidence for this is lacking. Furthermore, provided that a physical connection exists, it is unclear whether and how this translates into functional coupling between the two machineries.

Minor point: In Table S1 there seems to be a typo - multiplicity of the data for the Sm3+ structure 134.8?

Reviewer #2 (Remarks to the Author):

Through truncation constructs and limited proteolysis the authors identify short interacting regions of Rbm7, including the RRM, and a Pro-rich region of ZCCHC8.

Interestingly, the steric constraints imposed by the Pro residues of ZCCHC8 strongly influencing the interactions. The authors additionally identify potential interactions with pre-mRNA splicing factor SAP145, based on prior proteomic data, sequence similarities and in vitro binding.

Overall, the work is technically excellent and the MS is clearly written. Binding data are not quantified, but the results appear clear. The advance is modest, but makes a useful contribution to the field.

Minor point:

1) P9: immuniprecipitated => immunoprecipitated

We thank the Reviewers for their positive comments and their constructive criticisms. We have addressed the specific criticisms raised as outlined below.

Reviewer #1

1) Given the mutually exclusive binding of ZCCHC8 and SAP145 to RBM7, which might rather argue for an independent function of RBM7 in the two contexts, the authors speculate that dimerization of RBM7 might mediate a physical connection between the NEXT complex and the spliceosome (based on the known dimerization of a RBM7 paralogue, RBM11). However, direct evidence for this is lacking. Furthermore, provided that a physical connection exists, it is unclear whether and how this translates into functional coupling between the two machineries.

We agree with the Reviewer that RBM7 could have an independent function in the two processes in addition to the scenario we had proposed of linking the two processes via dimerization. We have modified this paragraph of the discussion accordingly.

Moreover, we have analyzed RNA-seq experiments employing HeLa cells subjected to RBM7 or ZCCHC8 depletion to monitor the effect of the NEXT complex on splicing. We found only a small set of altered splicing events, as shown in the **new Supplementary Figure 4**. Clearly, RBM7 and ZCCHC8 are not general splicing factors. Together with previous findings that RBM7 binds to a region in the 3' ends of introns, at a similar position where the U2snRNP binds (Lubas et al., 2015) and that SAP49 has been shown to crosslink near the branch site sequence in the splicing reaction (Champin Arnaud et al., 1994), the model that emerges is that the coupling of NEXT and SF3b might be a means to target intronic RNAs for degradation.

2) In Table S1 there seems to be a typo - multiplicity of the data for the Sm3+ structure 134.8?

The multiplicity is indeed correct. The crystals are in a hexagonal spacegroup and we collected a very redundant data set for the SAD experiment (1440° degree of total oscillation).

Reviewer #2

1) *Binding data are not quantified, but the results appear clear.*

We have now quantified the binding data by using microscale thermophoresis (MST) and found that RBM7^{R^{RM}} - ZCCHC8^{P^{ro}} interact with a K_D of 28nM (thus in the regime of the U2AF⁶⁵ - U2AF³⁵ⁱ interaction). These data are in the **new Figure 2H** of the revised manuscript.

2) *P9: immuniprecipitated => immunoprecipitated*

Corrected

Reviewers' Comments:

Reviewer #1 (Remarks to the Author):

In their revised version, the authors satisfactorily addressed all issues raised by this referee during the first round of reviews.

Reviewer #2 (Remarks to the Author):

The authors have addressed the minor points raised in my initial review and I am happy to recommend acceptance.

We thank the Reviewers for their positive comments and the suggestion to accept the manuscript.

Reviewer #1

In their revised version, the authors satisfactorily addressed all issues raised by this referee during the first round of reviews.

Reviewer #2

The authors have addressed the minor points raised in my initial review and I am happy to recommend acceptance.